# Coherent Signal DOA Estimation for MIMO Radar under Composite Background of Strong Interference and Non-Uniform Noise

**DOI:** 10.3390/s22249833

**Published:** 2022-12-14

**Authors:** Bin Lin, Guoping Hu, Hao Zhou, Guimei Zheng

**Affiliations:** 1Air and Missile Defense College, Air Force Engineering University, Xi’an 710051, China; 2Graduate College, Air Force Engineering University, Xi’an 710051, China

**Keywords:** non-uniform noise, strong interference signals, coherent signals, array signal processing, direction of arrival estimation

## Abstract

To address the problems of low accuracy and low robustness of the conventional algorithm in estimating the direction of arrival (DOA) of coherent signals against a composite background of strong interference and non-uniform noise, in this paper, a coherent signal DOA estimation algorithm based on fixed projection blocking is proposed in conjunction with a multi-input multi-output (MIMO) radar. The covariance matrix of the received signal is first decomposed by eigenvalues, and a fixed projection matrix orthogonal to the interference guidance vector is constructed as the interference blocking matrix. Then, the received array signal is pre-processed to re-form the covariance matrix, and this matrix is rendered decoherent through a Toeplitz reconstruction. Finally, the reconstructed covariance matrix is estimated by DOA using the propagation operator algorithm to reduce the complexity. The simulation verifies that the proposed algorithm has a better robustness and higher accuracy than conventional algorithms for the DOA estimation of coherent signals in composite backgrounds.

## 1. Introduction

The MIMO radar is a new radar system with many advantages over traditional phased array radars, including a low interception probability, strong anti-jamming capability and virtual arrays with large apertures. However, with the increasing complexity of the battlefield electromagnetic environment, MIMO radar detection also faces an increasing amount of interference, including coherent signals from multipath effects that interfere with each other, non-uniform noise caused by differences in antenna arrays and received signal channels, and strong interference signals from enemy jammers that are close to the radar or have a high effective interference power [1,2,3,4]. The effects of these disturbances will lead to problems of reduced accuracy, reduced stability and even failure of traditional direction-finding algorithms [5,6,7,8,9,10], which need to be addressed by effective technical means. If the sensor nodes can be connected to each other, the DV-hop algorithm proposed in the literature [11] can also help with localization.

In addition, to overcome the problem of a directional measurement under non-uniform noise, the authors of [12] pre-whitened received signal data to achieve a higher accuracy estimation, but the problem of aperture loss led to a limited number of measurement sources. The authors of [13] used the maximum likelihood estimation method to gradually iterate to obtain the signal subspace and noise covariance matrix, which improved the accuracy, because of the need for iterative operations; thus, the complexity was high. The authors of [14] used the centrosymmetric property of uniform line array to transform complex matrices into real-valued matrices and proposed the unitary low-rank matrix decomposition method to improve DOA estimation performance and reduce complexity. The authors of [15] constructed a spatial difference matrix to suppress noise and used a rotation-invariant subspace-like algorithm to solve the model sparsely, which has a strong robustness. To address the problem of coherent signal orientation under non-uniform noise, a generalized covariance matrix was constructed in [16] using a forward-backward spatial smoothing algorithm and the difference between the complex conjugate of the data matrix, which fully eliminated spatial non-uniform noise and achieved the estimation of coherent signals, but there was a problem of aperture loss. The authors of [17] proposed a DOA estimation based on the tail minimization method, which exploits the sparsity of the signal subspace and has the advantage of expanding the virtual array aperture. The authors of [18] used a spatial smoothing algorithm to reconstruct a full-rank signal covariance matrix, which is beneficial for the DOA estimation of uncorrelated or coherent signals, but with high complexity.

For the current problem of direction-finding in the context of strong interference, research ideas can be broadly divided into two types: one involves sieving out the strong interference signal and the weak signal after estimating them at the same time, and the other involves estimating the weak signal after removing the strong interference signal. For the former idea, the RELAX algorithm [19] method is mainly used; such an algorithm separates the array output part of all signals into multiple data blocks, and then, according to the characteristics of the interference signal, finds out the data blocks containing an interference signal and reject, so as to achieve the purpose of target identification; however, such an algorithm is problematic due to high complexity. The second idea is the subarray interference suppression algorithm proposed in the literature [20,21], which shows a good performance by improving the beam formation method, so that it can suppress interference before weighting the subarray for the DOA estimation of weak signals. The authors of [22] use the angular information of the signal matrix to construct an interference blocking matrix to suppress strong interference signals, but this method requires precisely known interference directions, and different blocking matrices need to be constructed for different array structures. Gong et al. [23] analyzed in detail the direction-preserving orthogonality in array signal data under the background of strong interference and performed simulations for a variety of scenarios, verifying that the algorithm was suitable for situations with low signal-to-noise ratios (SNR) and small angular spacing. The authors of [24] propose a broadband signal localization method based on sparse spectral fitting and a zero-value matrix filter with a high localization accuracy in a strong interference environment. For the DOA estimation of coherent sources in a strong interference background, Li et al. [25] proposed an improved interference blocking algorithm to render the coherent signal decoherent by Toeplitz matrix reconstruction, and then applied the interference blocking matrix method to suppress the interfering signal, but the algorithm was not robust.

The above algorithms suffer from a low estimation accuracy, poor robustness and high complexity in direction-finding against a composite background of strong interference and non-uniform noise. This paper presents a coherent weak-target DOA estimation method based on a fixed projection blocking algorithm and Toeplitz matrix reconstruction. The method obtains the interference subspace by decomposing the eigenvalues of the received signal covariance matrix, then constructs a fixed projection matrix orthogonal to the interference-oriented vector as the interference blocking matrix, multiplies it with the received array signal for pre-processing and re-forming the covariance matrix, performs a Toeplitz reconstruction to suppress non-uniform noise, and finally uses the propagator method (PM) for DOA estimation, avoiding the spectral peak search.

## 2. MIMO Radar Signal Model

As shown in Figure 1, it is assumed that the number of array elements of the transmitting array is *M*, the number of array elements of the receiving array is *N*, the array element spacing is *d*, the measured target is a far-field target, the signal reflected from the target is a narrow-band signal, and the wave front arriving at each array element is a plane wave.

*K* are far-field narrowband signals, and *J* are strong interference signals $s_{i}(t)(i=1,2,\ldots ,J,\ldots ,K+J)$ incident to the uniform line array; the first *J* signals are strong interference signals whose power satisfies $\sigma_{1}^{2}>\sigma_{2}^{2}>\cdots >\sigma_{J}^{2}$>>$\sigma_{J+1}^{2}>\cdots >\sigma_{K+J}^{2}$; the received *K* are far-field narrowband signals containing *Q* independent signals; *L* are the group of coherent signals in the presence of *P* coherent signals, where the number of coherent signals in the *l*-th group is $P_{l}$; the radar transmit and receive guide vectors are:(1)$$a_{t}\left(\theta \right)={\left[1,e^{-j2\pi d_{0}\sin\theta /\lambda },\cdots ,e^{-j2\pi \left(M-1\right)d_{0}\sin\theta /\lambda }\right]}^{T}$$
(2)$$a_{r}\left(\theta \right)={\left[1,e^{-j2\pi d_{0}\sin\theta /\lambda },\cdots ,e^{-j2\pi \left(N-1\right)d_{0}\sin\theta /\lambda }\right]}^{T}$$

Based on the orthogonality of the transmit waveform, the received signal at the *m*-th array element of the array at moment *t* after matching and filtering the signal received at the receiver end of the MIMO radar can be expressed as:(3)$$x_{m}(t)=\sum_{i=1}^{J}\left(a_{r}\left(\theta_{i}\right)⊗a_{t}\left(\theta_{i}\right)\right)s_{i}(t)+\sum_{k=1}^{K}\left(a_{r}\left(\theta_{k}\right)⊗a_{t}\left(\theta_{k}\right)\right)s_{k}(t)+n_{m}(t)$$
where $n_{m}(t)$ is the non-uniform noise of the *m*-th array element at moment *t*.

Rewriting the received signal into vector form, one obtains:(4)$$\begin{array}{c} x(t)=\sum_{j=1}^{J}\left(a_{r}\left(\theta_{j}\right)⊗a_{t}\left(\theta_{j}\right)\right)s_{j}(t)+\sum_{i=1}^{Q}\left(a_{r}\left(\theta_{i}\right)⊗a_{t}\left(\theta_{i}\right)\right)s_{i}(t)+\sum_{l=1}^{L}\sum_{k=1}^{P_{l}}\left(a_{r}\left(\theta_{lk}\right)⊗a_{t}\left(\theta_{lk}\right)\right)b_{lk}s_{c}(t)+n(t)\\ =A_{p}s_{p}(t)+A_{u}s_{u}(t)+A_{c}s_{c}(t)+n(t)\\ =As(t)+n(t) \end{array}$$
where $A_{p}=\left[a_{r}\left(\theta_{1}\right)⊗a_{t}\left(\theta_{1}\right),a_{r}\left(\theta_{2}\right)⊗a_{t}\left(\theta_{2}\right),\cdots ,a_{r}\left(\theta_{J}\right)⊗a_{t}\left(\theta_{J}\right)\right]$ denotes the array flow pattern of the strong interference signal, $s_{p}(t)={\left[s_{1}(t),s_{2}(t),\cdots ,s_{J}(t)\right]}^{T}$ denotes the incident signal vector of *J* strong interfering signals, $A_{u}=\left[a_{r}\left(\theta_{1}\right)⊗a_{t}\left(\theta_{1}\right),a_{r}\left(\theta_{2}\right)⊗a_{t}\left(\theta_{2}\right),\cdots ,a_{r}\left(\theta_{Q}\right)⊗a_{t}\left(\theta_{Q}\right)\right]$ indicates the array flow pattern of the independent signal, $s_{u}(t)={\left[s_{1}(t),s_{2}(t),\cdots ,s_{Q}(t)\right]}^{T}$ denotes the incident signal vector of *Q* independent signals, $A_{c}=\left[A_{c1}b_{1},\cdots ,A_{cL}b_{L}\right]$, $b_{l}={\left[b_{l1},b_{l2},\cdots ,b_{lP_{l}}\right]}^{T}$, $b_{lk}$ denotes the coherence coefficient of the *k*-th signal in the *l*-th set of coherent signals, $s_{c}(t)={\left[s_{J+Q+1}(t),s_{J+Q+2}(t),\cdots ,s_{J+Q+L}(t)\right]}^{T}$ denotes the incident signal vector of the *L*-group coherent signal, and $A_{cl}=\left[a_{r}\left(\theta_{l1}\right)⊗a_{t}\left(\theta_{l1}\right),a_{r}\left(\theta_{l2}\right)⊗a_{t}\left(\theta_{l2}\right),\cdots ,a_{r}\left(\theta_{lP_{l}}\right)⊗a_{t}\left(\theta_{lP_{l}}\right)\right]$, $A=\left[A_{p},A_{u},A_{c}\right]$, $s(t)={\left[s_{p}(t),s_{u}(t),s_{c}(t)\right]}^{T}$, n(t) is the non-uniform noise at the receiver.

The covariance matrix $R_{x}$ of the signal is expressed as:(5)$$R_{x}=E\left[x(t)x^{H}(t)\right]=AR_{s}A^{H}+R_{n}$$
where $R_{s}$ is the covariance matrix of the signal vector s(t), and $R_{n}=\mathrm{diag}\left\{\sigma_{1}^{\prime }{}^{2},\ldots ,\sigma_{MN}^{\prime }{}^{2}\right\}$ is the covariance matrix of the non-uniform noise n(t), where $\sigma_{1}^{\prime }{}^{2},\ldots ,\sigma_{MN}^{\prime }{}^{2}$ is the unequal noise power.

## 3. Coherent Signal DOA Estimation in A Composite Background

### 3.1. Fixed Projection Blocking Algorithm Rejects Interference

$R_{x}$ is the positive definite Hermitain matrix, for which the eigenvalue decomposition is:(6)$$R_{x}=U\Sigma U^{H}=\sum_{i=1}^{M}\lambda_{i}e_{i}e_{i}^{H}$$
where $U={\left[e_{1},e_{2},\cdots e_{MN}\right]}^{H}$, $\Sigma =\mathrm{diag}\left\{\lambda_{1},\lambda_{2},\ldots ,\lambda_{MN}\right\}$, $e_{i}$ is the eigenvector of matrix $R_{x}$, and $\lambda_{i}$ is the eigenvalue of matrix $R_{x}$, sorted as follows:(7)$$\lambda_{1}>\lambda_{2}>\cdots >\lambda_{J}>>\lambda_{J+1}>\cdots >\lambda_{J+K}>>\lambda_{J+K+1}>\cdots >\lambda_{\mathrm{MN}}$$

The eigenvalues are ordered to obtain the corresponding eigenvectors as $e_{1},e_{2},\cdots ,e_{MN}$, where the eigenvectors corresponding to the *J* large eigenvalues of the matrix $R_{x}$ form the strong interference subspace:(8)$$E_{J}=\left[e_{1},e_{2},\cdots ,e_{J}\right]$$

The eigenvectors corresponding to the *K* larger eigenvalues form the signal subspace:(9)$$E_{K}=\left[e_{J+1},e_{J+2},\cdots ,e_{J+K}\right]$$

Other eigenvectors corresponding to MN−K−J small eigenvalues form the noise subspace:(10)$$E_{N}=\left[e_{K+J+1},e_{K+J+2},\cdots ,e_{MN}\right]$$

Construct the fixed projection blocking matrix G:(11)$$G=I-E_{J}{\left(E_{J}^{H}E_{J}\right)}^{-1}E_{J}$$
which is orthogonal to the strong interference subspace, where I is the unit matrix of MN×MN dimensions.

The subspace $E=\left[E_{J},E_{K},E_{N}\right]$ is pre-transformed by the matrix G:(12)$$D=GE=G\left[E_{J},E_{K},E_{N}\right]=\left[0_{J},D_{J+K+1},D_{J+K+2},\cdots D_{MN}\right]$$

From the pre-transformation, it follows that the transformed subspace D is only relevant for the target signal and the noise, that is the fixed projection blocking matrix G is able to suppress strong interference signals.

Similarly, the data vector x(t) of the signal is subjected to interference rejection:(13)y(t)=Gx(t)

Subsequently, the new covariance matrix is obtained from the pre-processed signal data vector:(14)$$R_{y}=GAR_{s}A^{H}G^{H}+GR_{n}G^{H}$$

For the new covariance matrix, although the influence of interfering signals on the target signal is eliminated, the presence of coherent signals as well as non-uniform noise information in the matrix still affects the estimation accuracy of the signal.

### 3.2. Toeplitz Matrix Reconstruction

The matrix $R_{y}$ can be minimized by Toeplitzisation of the difference with the noiseless covariance matrix, that is:(15)$$\underset{R_{T}\in S_{T}}{\min}\left\|R_{T}-R\right\|$$
where $R_{T}$ is the Toeplitz reconstruction of $R_{y}$, R is the noiseless covariance matrix, and $S_{T}$ is the set of Toeplitz matrices. Essentially, the covariance matrix is averaged over the diagonal elements of the covariance matrix.

Define each element of the matrix as:(16)$$R_{y}=\left[\begin{array}{ccccc} r_{y}(1,1) & r_{y}(1,2) & r_{y}(1,3) & \cdots & r_{y}(1,MN)\\ r_{y}(2,1) & r_{y}(2,2) & r_{y}(2,3) & \cdots & r_{y}(2,MN)\\ \vdots & \vdots & \vdots & \vdots & \vdots \\ r_{y}(MN,1) & r_{y}(MN,2) & \cdots & \cdots & r_{y}(MN,MN) \end{array}\right]$$

Construct a new matrix by extracting the elements of the first row of the matrix $R_{y}$:(17)$${\tilde{R}}_{y}=\left[\begin{array}{ccccc} r_{y}(1,1) & r_{y}(1,2) & r_{y}(1,3) & \cdots & r_{y}(1,MN)\\ 0 & r_{y}(1,1) & r_{y}(1,2) & \cdots & r_{y}(1,MN-1)\\ 0 & 0 & r_{y}(1,1) & \cdots & r_{y}(1,MN-2)\\ \vdots & \vdots & \vdots & \vdots & \vdots \\ 0 & 0 & \cdots & \cdots & r_{y}(1,1) \end{array}\right]$$

The correlation vector of Equation (17) contains all the angular information of the incident source, and a Toeplitz matrix can be constructed from this vector with the following structure:(18)$${\tilde{\tilde{R}}}_{y}=\left[\begin{array}{ccccc} r_{y}(1,1) & r_{y}(1,2) & r_{y}(1,3) & \cdots & r_{y}(1,MN)\\ r_{y}^{*}(1,2) & r_{y}(1,1) & r_{y}(1,2) & \cdots & r_{y}(1,MN-1)\\ r_{y}^{*}(1,3) & r_{y}^{*}(1,2) & r_{y}(1,1) & \cdots & r_{y}(1,MN-2)\\ \vdots & \vdots & \vdots & \vdots & \vdots \\ r_{y}^{*}(1,MN) & r_{y}^{*}(1,MN-1) & \cdots & \cdots & r_{y}(1,1) \end{array}\right]$$
where ${\tilde{\tilde{R}}}_{y}$ is the MN×MN-dimensional Hermitian matrix. As the effect of non-stationary noise is concentrated in the main diagonal of the covariance matrix and the Toeplitz matrix is reconstructed with the main diagonal elements of the covariance matrix averaged, the effect of non-stationary noise is reduced and decoherence is achieved, but the matrix also suffers from excessive complexity compared to conventional algorithms.

### 3.3. PM Algorithm DOA Estimation

According to the literature [26], the matrix ${\tilde{\tilde{R}}}_{y}$ is divided into two submatrices:(19)$${\tilde{\tilde{R}}}_{y}={\left[R_{K1}^{T},R_{K2}^{T}\right]}^{T}$$
where $R_{K1}$ consists of the first *K* rows of the matrix ${\tilde{\tilde{R}}}_{y}$ (*K* is the number of target signals), and $R_{K2}$ consists of the next MN−K rows. Since $R_{K1}$, $R_{K2}$ are full rank vandermond matrices, there exists an K×(MN−K)-dimensional linear operator ***P***, so that:(20)$$P={\left(R_{K1}R_{K1}^{H}\right)}^{-1}R_{K1}R_{K2}^{H}$$

For ease of calculation, define $\Omega ={\left[P^{T},-I_{MN-K}\right]}^{T}$ ($I_{MN-K}$ is the MN−K-dimensional unit matrix). Similarly, define $a(\theta )=a_{r}(\theta )⊗a_{t}(\theta )$, and the estimate θ of $\bar{\theta }$ is the value taken when minimizing the cost function f(θ):(21)$$f(\theta )=a^{H}(\theta )\Pi a(\theta )$$
where $\Pi =\Omega {\left(\Omega^{H}\Omega \right)}^{-1}\Omega^{H}$.

## 4. Basic Steps of the Algorithm

Based on the above analysis, the operational steps of the Toeplitz reconstruction algorithm based on fixed projection blocking can be summarized as follows:
Calculate the data covariance matrix R of the array element output vector x(t);The covariance matrix $R_{x}$ is decomposed to obtain the interference subspace $E_{J}$, the signal subspace $E_{K}$, and the noise subspace $E_{N}$;To eliminate interference, construct a fixed projection blocking matrix G using Equation (11);Pre-processing the received signal by Equation (13) gives y(t);The covariance matrix $R_{y}$ is reconstructed according to Equation (18) and replaced to obtain the matrix ${\tilde{\tilde{R}}}_{y}$;After partitioning the matrix ${\tilde{\tilde{R}}}_{y}$, construct the linear operator P according to Equation (20);The DOA is obtained by minimizing the cost function x using Equation (21).

## 5. Algorithm Performance Analysis

### 5.1. Source Overload Capacity Analysis

For transceiver arrays consisting of *M* and *N* arrays, the proposed algorithm is able to identify MN−1 coherent signals without loss of aperture using the Toeplitz matrix reconstruction algorithm. If the forward spatial smoothing algorithm is used to identify MN/2 coherent signals at most, the forward and backward spatial smoothing algorithm can identify 2MN/3 coherent signals at most. Compared with the conventional algorithm, the proposed algorithm has an advantage in the number of coherent sources that are identified.

### 5.2. Algorithmic Complexity Analysis

Assuming that the number of sources measured is *K*, the number of samples is *L*, and the number of angular searches is $N_{\theta }$, the complexity of the fixed projection blocking-Toeplitz (FPB-To) algorithm proposed in this section is mainly derived from the covariance matrix formation, the eigenvalue decomposition and the PM algorithm, and the computational effort is approximated by $O\left(2L{\left(MN\right)}^{2}+{\left(MN\right)}^{3}+10K^{3}+4KMN\right)$. In the case of $N_{\theta }>L>MN>K$, the comparison is slightly lower than for the jamming jam method (JJM) algorithm, the extended noise subspace (ENS) algorithm, the Toeplitz-jamming jam method (To-JJM) algorithm [25] with complexity $O\left(L{\left(MN\right)}^{2}+{\left(MN\right)}^{3}+N_{\theta }{\left(MN\right)}^{2}\right)$, and the fixed projection blocking (FPB) algorithm with complexity $O\left(2L{\left(MN\right)}^{2}+2{\left(MN\right)}^{3}+N_{\theta }{\left(MN\right)}^{2}\right)$.

## 6. Simulation Results and Analysis

The simulation uses a uniform line array with the number of array elements *M* and *N*. $R_{n}=\mathrm{diag}\left\{\sigma_{1}^{\prime }{}^{2},\ldots \sigma_{MN}^{\prime }{}^{2}\right\}$, $\sigma_{m}^{\prime }{}^{2}(m=1,\ldots ,MN)$ is the different noise power of each array element; then, the signal-to-noise ratio is defined as $\mathrm{SNR}=10\log_{10}(\frac{1}{MN}\sum_{m=1}^{MN}\frac{\sigma_{k}^{2}}{\sigma_{m}^{\prime }{}^{2}})$, and the interference signal ratio (ISR) is defined as $10\log_{10}\left(\sigma_{j}^{2}/\sigma_{k}^{2}\right)$, where $\sigma_{k}^{2}$ is the average power of the target signal and $\sigma_{j}^{2}$ is the average power of the strong interference signal.

The root mean square error (RMSE) of the DOA estimate is:(22)$$\mathrm{RMSE}=\sqrt{1/2N_{mont}\sum_{i=1}^{N_{mont}}\sum_{k=1}^{K}\left({\left|\theta_{i}-\theta_{ik}\right|}^{2}\right)}$$
where $N_{mont}$ is the number of Monte–Carlo experiments, *K* is the number of signals, and $\theta_{i}$ and $\theta_{ik}$ are the true and estimated values of the *i*-th signal angle in each trial, respectively.

### 6.1. Spatial Spectrum Estimation

The simulation sets *M* = *N* = 12; there is one set of coherent signals with a coherence coefficient of 0.9, incidence angles of −60° and –40°, three non-coherent signals with incidence angles of –20°, 0° and 20°, and two strong interfering signals with incidence angles of 40° and 60°; the SNR is 0 dB, the ISR is 30 dB, and the number of snapshots is 350. Figure 2 gives the spatial spectral curves of the JJM algorithm, ENS algorithm, FPB algorithm, To-JJM algorithm and the spatial spectrum curve of the FPB-To algorithm proposed in this paper.

Figure 2 shows that the JJM algorithm, ENS algorithm, and FPB algorithm all have specific interference cancellation mechanisms, which can suppress strong interference signals and estimate the DOA of non-coherent signals better, but they are less effective in estimating coherent signals. The To-JJM algorithm can better estimate the DOA of coherent targets in a composite background but requires an accurate prediction of the direction of strong interference signals, which is not conducive to the engineering application of the algorithm. The FPB-To algorithm is the most effective and can accurately detect the target DOA without predicting the direction of interference.

### 6.2. Comparative Error Analysis of Different Algorithms

The number of Monte–Carlo experiments performed is 500, and the other settings are the same as in Section 6.1. Figure 3a provides the curves of RMSE versus snapshots for the JJM algorithm, ENS algorithm, FPB algorithm, To-JJM algorithm, and FPB-To algorithm by varying the snapshots from 50 to 500 in steps of 50. Figure 3b shows the curve of RMSE with SNR for the five algorithms by varying the SNR from –10 dB to 10 dB in steps of 2 dB.

As can be seen from Figure 3, the RMSE of these methods decreases as the SNR and the number of snapshots increase. As the number of snapshots and the SNR increase to a certain level, the RMSE of several algorithms level off. The To-JJM algorithm outperforms several other algorithms at a higher SNR due to the decoherence of the target and non-uniform noise suppression. The algorithm in this paper is pre-transformed in the subspace, which reduces the sensitivity to noise to a certain extent and still provides a good directional accuracy at a low SNR.

### 6.3. Comparative Error Analysis with Different Numbers of Interference Sources

We change the number of interference sources on the basis of the settings in Section 6.1. For a strong interference number of 1, the angle of incidence is –40°; for a strong interference number of 2, the angles of incidence are –40° and –60°; for a strong interference number of 3, the angles of incidence are –40°, –60°, and –80°. Figure 4a shows the curve of the To-JJM algorithm SNR varying from –10 dB to 10 dB in steps of 2 dB, with RMSE varying with SNR for three numbers of interference sources. Figure 4b shows the curve of RMSE versus SNR for the FPB-To algorithm SNR varying from –10 dB to 10 dB in steps of 2 dB for three sources of interference.

As can be seen from Figure 4, both algorithms still do a good job of suppressing interference and accurately estimating weak signals against the composite background. The To-JJM algorithm performs better when suppressing one interfering signal than when suppressing three, the reason being that for every interfering signal suppressed by the algorithm, a certain amount of aperture loss is incurred, which affects the direction-finding accuracy. However, the proposed FPB-To algorithm does not make a significant difference as the number of interference sources increases, and still achieves a high accuracy in direction finding.

### 6.4. Algorithm Robustness Analysis

To measure the difference in noise intensity received in different arrays, the worst noise power ratio (WNPR) is defined as:(23)$$\mathrm{WNPR}=\sigma_{m}^{\prime }{}_{\max}^{2}/\sigma_{m}^{\prime }{}_{\min}^{2}$$

This is the ratio of the power of the strongest noise to the weakest noise received by each physical array element.

The simulation setup is the same as in Section 6.1. Figure 5a provides the curves of the algorithm WNPR varying from 10 to 100 in steps of 10, with the RMSE varying with WNPR for different algorithms.

In addition, the simulation is set up with three different noise covariance matrices:(24)$$\begin{array}{l} N1=\mathrm{diag}\{1,1,1,2,2,2,2,3,3,3\}\\ N2=\mathrm{diag}\{5,9,15,2,3,4,2,4,4,1\}\\ N3=\mathrm{diag}\{1,2,3,4,5,1,2,3,4,5\} \end{array}$$

Other simulation conditions are the same as in Section 6.1. Figure 5b shows the spatial spectral estimation performance of the proposed algorithm for different noise covariance matrices.

As can be seen from Figure 5, the WNPR characterizes the inhomogeneity of the non-uniform noise, and the larger the WNPR, the stronger the inhomogeneity; additionally, the algorithm performances are all degraded to different degrees. Compared with other algorithms, the algorithm in this section is able to maintain an estimation accuracy under different WNPRs and noise covariance matrices, and has a strong robustness.

## 7. Conclusions

In answer to the problem of coherent source vectoring in the composite background of strong interference and non-uniform noise, this paper proposes a fixed projection blocking algorithm based on Toeplitz reconstruction. The algorithm first constructs the covariance matrix from the received signal vector, while an eigenvalue decomposition is then performed, the resulting strong interference vector is used to construct the orthogonal projection matrix of the interference subspace, and the covariance matrix is obtained again after pre-processing the received array signal. This is followed by a Toeplitz matrix reconstruction for decoherence and finally a PM algorithm to obtain the DOA information of the faint signal. The experimental results show that the proposed algorithm is able to perform a DOA estimation of coherent targets in strong interference and non-uniform noise environments with a higher accuracy and robustness than conventional algorithms. It is conducive to a better use in future complex and changing electromagnetic environments [27,28,29,30,31].

## Figures and Tables

**Figure 1 sensors-22-09833-f001:**
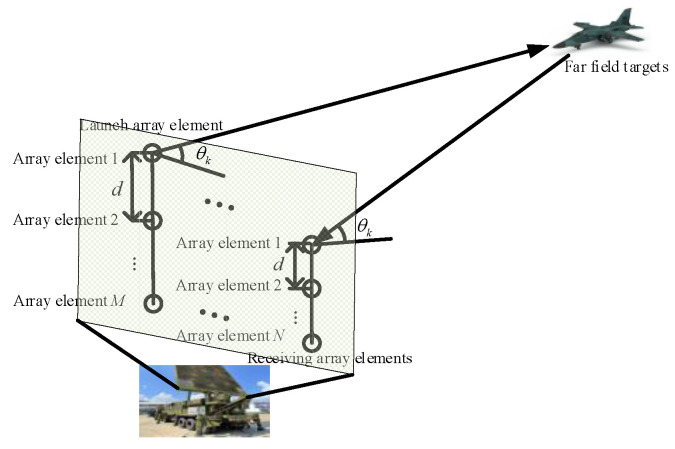
Schematic diagram of the MIMO radar architecture.

**Figure 2 sensors-22-09833-f002:**
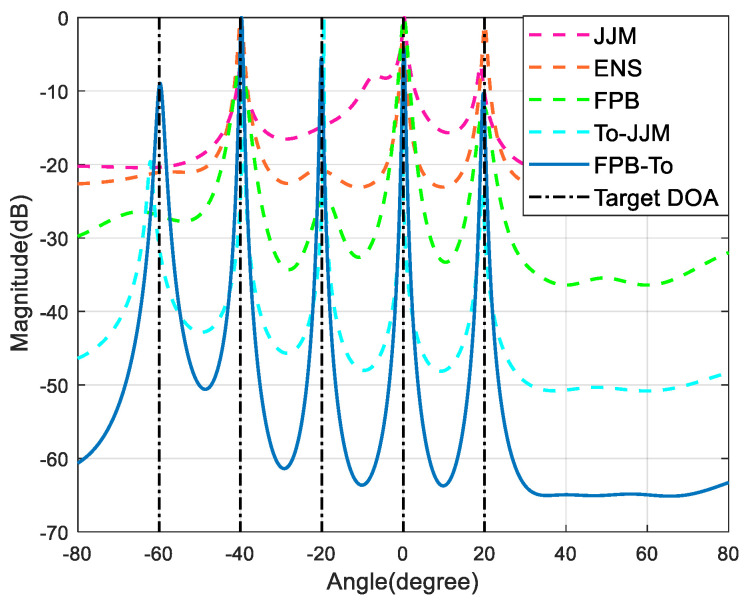
Comparison of spatial spectral curves.

**Figure 3 sensors-22-09833-f003:**
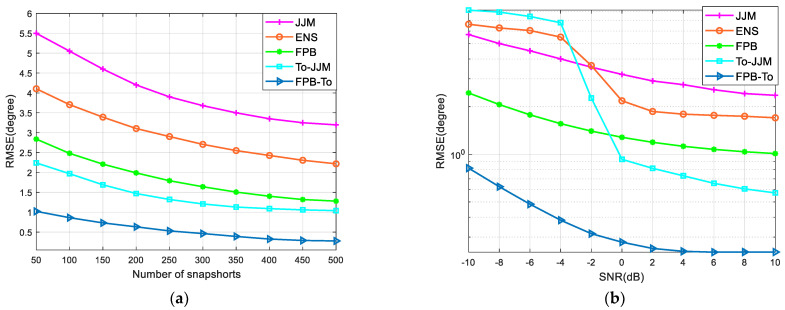
RMSE analysis: (**a**) RMSE in relation to snapshots; (**b**) RMSE in relation to SNR.

**Figure 4 sensors-22-09833-f004:**
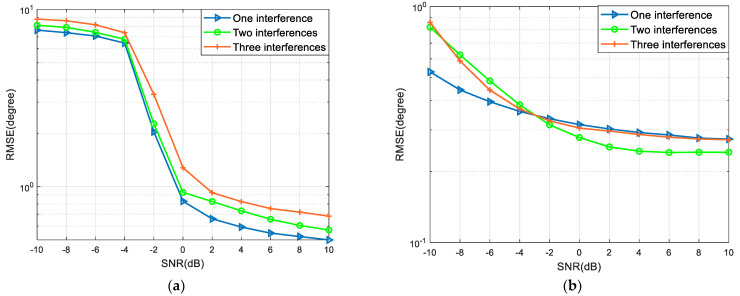
Variation curve of algorithm performance with SNR under different numbers of interference sources: (**a**) To-JJM algorithm; (**b**) FPB-To algorithm.

**Figure 5 sensors-22-09833-f005:**
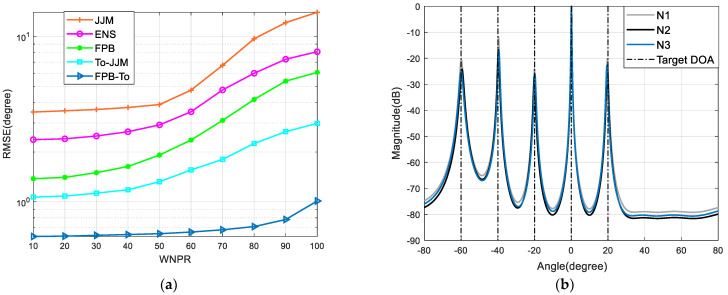
Algorithm robustness analysis: (**a**) Comparison of errors at different WNPRs; (**b**) Spatial spectral curves for different noise parameters.

## Data Availability

The data presented in this study are available on request from the corresponding author. The data are not publicly available, due to the data in this paper not coming from publicly available datasets but being obtained from the simulation of the signal models listed in the article.

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
