# Peer review of "Coherent Signal DOA Estimation for MIMO Radar under Composite Background of Strong Interference and Non-Uniform Noise"

_sensors, 2022, doi:10.3390/s22249833_

Round 1
Reviewer 1 Report
1 “one is to sieve out the strong interference signal and the weak signal after estimating them at the same time, and the other is to estimate the weak signal after removing the strong interference signal. ”The correspond sequence with following context is uncorrect. For example CLEAN belogs to the second type.
2 How about the performance of figure 2 at different snapshots, or at different numbers of coherent signals?
3 How to set 3 different noise covariance matrices?
4 Toeplitz Matrix Reconstruction to solve DOA problem of coherent signals is a common method. What is the differences between the proposed method and conventional methods?
5 How to estimate the "strong" interference signal? for example, what about the threshold?
Reviewer 2 Report
This paper proposed a coherent signal DOA estimation algorithm based on fixed projection blocking, to improve the accuracy and robustness of DOA estimation with MIMO radar. The simulation showed the better performance of the proposed algorithm in DOA estimation of coherent signals than some existing algorithms. The research topic is important and should be very interesting to the subscribers of this journal. Nevertheless, there are some minor issues to be addressed.
First, at the beginning of the introduction, this paper mentioned the "increasing complexity of the battlefield electromagnetic environment". In this complex environment, what happens if radars cannot work? The authors are suggested to introduce a bit about range-free localization method, such as the article titled Connectivity Based DV-Hop Localization for Internet of Things published in ieee transactions on vehicular technology. Even in a complex environment where the radars cannot work, DV-hop algorithm can help to localize sensor nodes as long as they can connect with each other.
Second, in the introduction, it would be better if the authors can explain the advantage of MIMO radar compared with other radars. Or the authors can explain why they focus on MIMO radar rather than other radars.
Third, "4. Basic Steps of the Algorithm and Complexity Analysis", in this section, it seems that there is no complexity analysis. So the title of this section should be corrected.
Round 2
Reviewer 1 Report
1) I am afraid the kernel of decoherence by Toeplitz Matrix
reduce the number of samples. Maybe different row vector of matrix will bring different results. And the kernel step (18) is the common decoherence step as the same as conventional methods. Therefore, I am afraid the kernel to manage decoherence is (18) not Toeplitz reconstruction. Maybe more numerical analyses should be given to test these judgements.
2) The variables like d(row 183) lack explanations.
